# Alternative Molecular Tools for the Fight against Infectious Diseases of Small Ruminants: Native Sicilian Sheep Breeds and Maedi-Visna Genetic Susceptibility

**DOI:** 10.3390/ani12131630

**Published:** 2022-06-24

**Authors:** Serena Tumino, Marco Tolone, Paola Galluzzo, Sergio Migliore, Tiziana Sechi, Salvatore Bordonaro, Roberto Puleio, Antonello Carta, Guido Ruggero Loria

**Affiliations:** 1Department of Agriculture, Food and Environment, University of Catania, Via Valdisavoia 5, 95123 Catania, Italy; serena.tumino@unict.it; 2Department of Agricultural, Food and Forest Sciences, University of Palermo, 90128 Palermo, Italy; marco.tolone@unipa.it; 3OIE Reference Laboratory for Contagious Agalactia, Istituto Zooprofilattico Sperimentale della Sicilia, 90129 Palermo, Italy; paola.galluzzo@izssicilia.it (P.G.); roberto.puleio@izssicilia.it (R.P.); guidoruggero.loria@izssicilia.it (G.R.L.); 4Genetics and Biotechnology–Agris Sardegna, 07040 Olmedo, Italy; tsechi@agrisricerca.it (T.S.); acarta@agrisricerca.it (A.C.)

**Keywords:** *TMEM154* gene, maedi-visna, susceptibility, autochthonous sheep

## Abstract

**Simple Summary:**

Local breeds represent a precious reservoir of genetic diversity, crucial to adapting to environmental and climate changes and reacting to evolving diseases. In Sicily, four native dairy breeds, namely Valle del Belìce, Comisana, Barbaresca, and Pinzirita, have adapted to low-input farming systems and semiarid environments, having an essential role in producing high-quality milk and typical dairy products. Maedi-visna (MV) is one of the most important chronic diseases affecting the sheep sector worldwide, causing production losses. Different target genes play an important role in immunity and in genetic resilience to MV, such as *TMEM154*, *TLR9*, *MYD88*, and CCR5. A major host genetic component to sheep MV susceptibility was identified in the ovine *TMEM154* gene. Animals with either of TMEM154 haplotypes that encode glutamate at position 35 (E35) of the protein are at higher risk of MV infection than those homozygous with lysine at position 35 (K35). In the tested Sicilian breeds, animals carrying the allele E35 showed a greater risk of being serologically positive. Comisana, Barbaresca, and Pinzirita breeds showed a good frequency of the protective allele K35, whilst a high frequency of risk allele was found in the Valle del Belìce breed, related to the selection strategies addressed to obtain a productive dairy sheep. Our results highlight the importance of the preservation of autochthonous breeds as a reservoir of natural resistance against infectious disease.

**Abstract:**

Maedi-visna (MV) is a disease caused by small ruminant lentiviruses. It is included in the list of notifiable terrestrial animal diseases due to economic losses and animal welfare harm in the sheep sector. To date, control programs remain the onliest approach to avoiding infection. The allelic variant p.Glu35Lys (E35K) of the *TMEM154* gene has been strongly associated with host vulnerability to MV illness. The present study aimed to investigate the association of *TMEM154* E35K allele frequencies with MV susceptibility in native Sicilian sheep breeds. More than 400 animals from 14 local sheep were serologically tested and genotyped for the *TMEM154* E35K polymorphism. The local breeds displayed different values of MV seroprevalence, with the lowest antibody prevalence in Barbaresca and Pinzirita breeds. *TMEM154* protective allele (K35) was less frequent than the risk allele (E35) in Valle del Belìce breed, whereas the other three breeds showed a more balanced alleles distribution. A positive association between seroprevalence and genotype was found in the entire sample set. The risk of infection resulted in more than 3-fold times as high in sheep with EK and EE genotype compared to the KK genotype. Our data could be helpful in establishing selection breeding programs aimed at reducing MV infection in Sicilian sheep farming and encouraging the breeding of native breeds.

## 1. Introduction

Maedi-visna (MV) and caprine arthritis/encephalitis (CAE) are viral diseases caused by two closely related *Lentiviruses* belonging to the Retroviridae family that occur in sheep and goats, respectively, with severe economic losses to the small ruminant sector worldwide [1]. The phylogenetic correlation between MV virus (MVV) and CAE virus (CAEV) shows explicit evidence of the existence of cross-species transmission between sheep and goats and, for this reason, these strains have been generally called small ruminant lentiviruses (SRLV) [2,3,4,5].

Both diseases are characterized by a prolonged incubation period and slow progressive subclinical infection, consisting of a chronic interstitial infiltration of mononuclear inflammatory cells in several organs, such as the lung, mammary gland, joints, and central nervous system. The clinical signs depend on the tropism of the strain, and they are generally mild or subclinical and related to the host genetic susceptibility and organ involved. In Mediterranean sheep and goats, generally, the disease causes a constant decrease in milk production accompanied by sclerosis of the udder nevertheless MV may evolve to respiratory or neurologic syndromes leading to progressive physical decay, cachexia, and death [1,6,7,8,9,10].

The primary route of infection is related to ingestion of infected colostrum, milk, or also by exposure to respiratory secretions (horizontal transmission). Vertical transmission (transplacental) and via semen [11] represent alternative routes of infection.

SRLV have been included by the World Organization for Animal Health (OIE) in the list of notifiable terrestrial animal diseases due to their economic impact on the international trade of animals and their products [1,10].

Like other viral diseases, there is no treatment against SRLV infections. The SRLV’s genetic variability, constantly enlarged by descriptions of new circulating genotypes and subtypes [12,13,14], contributes to making all efforts for the development of vaccines inconclusive. The eradication programs are expensive but remain the foremost sustainable approach to reducing SRLV infection in the small ruminant industry. 

Different candidate loci have been previously reported to be involved in MV infection with various levels of statistical significance [15,16,17,18]; however, a genome-wide association study in North American sheep identified a single, major gene (*transmembrane protein 154* gene, *TMEM154*) significantly associated with host vulnerability to MV illness [19].

Among the different haplotypes described by Heaton et al. [19], haplotypes 1, 2, and 3 have been the most frequent ones, distributed worldwide. Haplotype 2 or 3, both of which encode a glutamate amino acid residue at position 35 (E35) of the extracellular portion of TMEM154 protein, have been associated with an increased risk of MV infection. Conversely, ewes homozygous for haplotype 1, which encodes a lysine residue at position 35 (K35), have been strongly associated with being less likely to become infected [19,20,21,22,23,24,25,26,27].

In Sicily, the ovine dairy sector represents a significant economic income, particularly for those marginal rural areas in which the harsh environment limits alternative economic activities. After Sardinia, Sicily is the second Italian region for the number of sheep reared for milk production, counting about 699,000 animals (data provided by BDN–Anagrafe Zootecnica). In Sicily, there are four native dairy breeds: Valle del Belìce (VDB), Comisana (COM), Barbaresca (BAR), and Pinzirita (PIN), which represent a unique example of our livestock biodiversity. These breeds have an essential role in producing high-quality milk and dairy products although they survive in often difficult climatic conditions and semiarid environments.

The most common sheep breed reared in Sicily is Valle del Belìce, which originates in 1980 from hybridization among the native Pinzirita and Comisana breeds and Sarda rams. The selective interbreeding has given rise to a new biotype of sheep, which combines the characteristics of the three original breeds resulting in the highest productive and rustic animal for dairy farming [28].

Comisana is a multi-attitude breed originating from sheep breeds of the Southern Mediterranean area, and its name came from the old village called “Comiso”, located in a livestock area of Ragusa Province. 

Barbaresca is an ancient Sicilian fat-tail sheep that originated from crossbreeding between Tunisian Barbary rams from North Africa and the local Pinzirita breed [29].

Pinzirita, small size ewe adapted to graze in hilly areas, originates from the Asian or Syrian sheep of the Sanson, precisely from the Zackel strain, probably the most ancient breed of the group [30].

Because there is no information about genetic susceptibility to MV in the Sicilian sheep breeds, the present study was aimed to investigate the association of p.Glu35Lys (E35K) polymorphism of the *transmembrane protein 154* (*TMEM154*) gene with the serological MV status of tested animals to estimate the effect of the risk allele (E35) on the hazard of MV infection and in order to provide data about the frequencies of the protective allele (K35) in the local sheep breeds.

## 2. Materials and Methods

### 2.1. Sampling

The study sampling was carried out in the selected farms, during the official controls for Brucella National Plan: two vials of blood were collected from the jugular vein with vacuum blood tubes with and without EDTA. Blood samples were immediately taken to the laboratory and stored at 4 °C until use. All the procedures were in agreement with the recommendations of the European Union Directive 2010/63/EU, to respect animal welfare. 

A total of 442 animals (70 rams, 372 ewes) from 14 flocks, across 6 provinces (Agrigento, Enna, Caltanissetta, Catania, Palermo, Trapani) of Sicily (South Italy) were collected between 2019 and 2021. All sampled animals were over three years old and belonged to the following groups of sampling: Valle del Belìce (VDB), Comisana (COM), Barbaresca (BAR), and Pinzirita (PIN) (details on numbers of flocks, sheep per breed and per flock, and province are given in Table 1). In the screened flocks, no anamnestic data on MV were reported, nor clinical signs of MV disease. 

### 2.2. Serological Testing for MV Status

The serum was separated by centrifugation at 3000× *g* for 10 min and stored at −20 °C. Serum samples were processed at the laboratory of the Virological Diagnostic Area (Istituto Zooprofilattico Sperimentale della Sicilia). Serological tests were performed by a commercial enzyme-linked immunosorbent assay (ELISA) (CAEV/MVV^®^ELISA-IDEXX Laboratories, Liebefeld, Switzerland), according to the manufacturer’s instructions.

The cut-off value was defined based on the corrected optical density (OD) at a wavelength of 450 nm ratio of sample to a positive control (S/P). According to the manufacturer’s instructions, samples are considered SRLV negative with an S/P value ≤ 110% and SRLV positive with an S/P value ≥ 120%. Doubtful results, given with a value in the range of 110–120%, were excluded from further analysis. Serological positive individuals were tested twice to exclude false positives within samples.

### 2.3. DNA Extraction and TMEM154 Genotyping

Genomic DNA was obtained from whole blood using the Quick-DNA™ Miniprep Plus Kit (Zymo, Irvine, CA, USA), following the protocol provided by the manufacturer.

DNA quantity and quality were measured by a Nanodrop ND-1000 spectrophotometer (Thermo Fisher Scientific Inc., Waltham, MA, USA). 

The *TMEM154* genotyping was assessed by a TaqMan allelic discrimination assay using primers and probes set according to Ramìrez et al. [26]. 

The TaqMan allelic discrimination assay was performed using the Bio-Rad CFX96 Real-Time PCR Detection System in a total volume of 20 µL containing 1X SsoAdvancedTM Probes^®^ Supermix (Bio-Rad Laboratories, Hercules, CA, USA), 500 nM of primers, 250 nM of each probe, and ~150 ng of DNA. The real-time PCR program consisted of a DNA polymerase activation and DNA denaturation step (3 min at 95 °C), followed by 40 amplification cycles of denaturation for 20 s at 95 °C and annealing-elongation for 30 s at 60 °C. Each assay included positive controls representing all polymorphism combinations and no-template control (NTC) in triplicates. Genotypes of unknown samples were considered reliable if the results for all reference samples were correct. 

The accuracy and robustness of the TaqMan allelic discrimination assay were tested by comparing it with the results of direct Sanger sequencing. 

Therefore, a region including exons 2 and 3 of the *TMEM154* gene was amplified according to Heaton et al. [19]. PCR products (771 bp) were purified by Exo-SAP incubation followed by chain termination reaction with the BigDye™ Terminator v3.1 Cycle Sequencing Kit (Thermo Fisher Scientific Inc., Waltham, MA, USA). Sequencing reactions were performed sequentially on Applied Biosystems 3500 genetic analyzer (Applied Biosystem, Foster City, CA, USA). All samples were sequenced bidirectionally. Furthermore, Sanger sequencing was also aimed to check the presence of potential new SNPs in the Sicilian breeds that could have interfered with the TaqMan allelic discrimination probes. Primer and probes sequences are listed in Table 2.

### 2.4. Statistical Analysis

Chromatograms were visualized using BioEdit software (Tom Hall, Ibis Biosciences, Carlsbad, CA, USA) and aligned by MEGA v6.0 software (Pennsylvania State University, State College, PA 16801, United States) [31]. The R statistical software v. 3.6.2 was used for the following statistical analyses. Genotypic and allelic frequencies were calculated, and the chi-squared test was used to test whether the population deviated from the Hardy–Weinberg equilibrium (HWE).

Association analysis using the chi-square test was only performed for breeds that had a prevalence of serological positive between 10% and 90% and included individuals carrying protective genotype (KK) as well as non-protective genotypes (EK or EE).

The relative risk (RR) to be serologically MV positive (in an MV-affected flock) was estimated for animals carrying one and/or two copies of the susceptible allele (risk factor) with the method of Altman et al. [32] using the following equation:$$\mathrm{RR}\;=\;\frac{a/\left(a+b\right)}{c/\left(c+d\right)}$$
where *a* is the number of serologically MV-positive individuals carrying the risk factor, b is the number of serologically MV-negative individuals carrying the risk factor, *c* is the number of serologically MV-positive individuals carrying no risk factor, and *d* is the number of serologically MV-negative individuals carrying no risk factor.

## 3. Results

### 3.1. Serological MV Status of Sampled Breeds and Flocks

A total of 442 individuals belonging to four Sicilian sheep breeds, originating from 14 flocks, were sampled and tested for MV infection. ELISA method showed a seroprevalence of 26% (*n* = 117). The four breeds displayed different values of antibody prevalence: COM breed showed the highest seroprevalence (45%), VDB breed showed moderate seroprevalence (29%), whereas BAR and PIN showed the lowest values (4% and 1%, respectively).

MV seroprevalence ranged from 5% to 65%, considering flocks individually. Samples with serologically MV positive were identified in three out of four VDB flocks, one of which was among those with the highest seroprevalence (48%), three out of four farms of COM breed were seronegative flocks, the fourth one showed the highest seroprevalence (65%), whilst the lowest MV seroprevalence was observed in BAR and PIN flocks with 6% and 5%, respectively. Details on serological MV status within Sicilian flocks are given in Table 1.

### 3.2. TMEM154 Genotyping and Association Analyses

No new SNPs were found in the target *TMEM154* region, assessed by sequencing analysis. Therefore, all sampled animals have successfully genotyped for E35K polymorphism of the transmembrane protein 154 (*TMEM154*) gene using the TaqMan allelic discrimination high-throughput method. 

In the total sample set of 442 animals from 14 flocks, the putative protective allele (K35) at amino acid position 35 of TMEM154 protein was observed with a frequency of 24%. Within breed groups, the VDB breed is characterized by an unbalance of allelic and genotype frequencies but the number of sheep positive to serological assays is quite high, differently from BAR and PIN. Deviation from the Hardy–Weinberg equilibrium was only observed considering the total sample (*p*-value < 0.001) (Table 3). 

The association analyses were limited to the total sample set (442 samples) and COM breed (92 samples), but a significant association value (*p*-value <0.05) between *TMEM154* genotypes and serological MV status (positive vs. negative) was exclusively found in the total sample (Table 4). VDB, BAR, and PIN breeds were excluded from association analyses due to the low prevalence of MV-positive sheep (less than 10%) and/or inadequate balance of *TMEM154* genotype frequencies. 

The relative risk (RR) to be serologically MV positive was calculated for sheep carrying one or two copies of the putative risk factor (E35 allele) compared to animals carrying no risk allele. In the VDB, BAR, and PIN subgroups, the absence of serologically MV-positive sheep with KK genotype produced large confidence intervals. Therefore, this analysis was only performed considering the total set sample and the COM breed subset. In the total sample set and in the COM breed subgroup, the relative risk of being seropositive ranged from 2.5 to 3.8 times as high in sheep carrying the genotypes containing the risk allele (E35), namely animals with EK and EE genotype, compared to sheep with the KK genotype. However, a significant difference in the relative risk to be serologically MV positive for sheep carrying the EE and EK genotypes compared to sheep with the KK genotype was only observed for the total sample set (Table 4). 

## 4. Discussion

SRLV infection is widespread across the world, its incidence is globally differentiated with the highest individual prevalence in Europe [33], which seems to be related to the management system and inefficient controls on the imports of dairy small ruminants. To date, there is neither an effective vaccine nor treatment against MV disease; therefore, control programs remain the onliest approach to avoiding infection. The scarcity and patchy of epidemiological data represent a significant obstacle to implementing common MV disease eradication strategies that are currently heterogeneous and applied at local levels [1,10,33]. Furthermore, all costs linked to some eradication programs, such as culling or artificial rearing of lambs and maintenance of separate enclosures for seropositive and seronegative animals, are noteworthy and may be impractical. Selection and maintenance of traits that confer resistance to infectious disease are promising ways to complement existing control measures. The existence of genetic variation in resistance to various major endemic diseases in sheep has been highlighted [34], and one of the most well-known examples of marked assisted selection (MAS) is the case of the strong resistance to scrapie, conferred by specific genotypes at the prion gene (PRNP) [35].

Ovine transmembrane protein 154 gene (*TMEM154*) has been shown to play a central role in susceptibility to MV infection. Haplotypes encoding a glutamate amino acid residue at position 35 (E35) have a greater risk of MV infection than sheep with haplotypes carrying a lysine residue (K35) [19]. 

Multiple observations about the strong evidence for this association were observed in American, German, Iranian, Turkish, Spanish, and Italian sheep breeds [21,22,23,24,25,26,36].

Furthermore, in a recent study, Murphy et al. [27] highlighted the strong evidence for lifetime resiliency to SRLV infection for ewe carrying the haplotype 1 (K35 allele), correspondingly, their improved productivity in terms of the total number and weight of lamb weaned. Therefore, *TMEM154* genotyping stands as the most promising candidate in genetic selection against MV infection. Reducing the frequencies of susceptible genotypes (EE/EK) by selective breeding programs would enhance the effectiveness of test and cull strategies. 

The main goal of the present study was to investigate the association of *TMEM154* E35K allele frequencies with MV susceptibility in native Sicilian sheep breeds. The seroprevalence at the flock level ranged between 5 and 65%, and in general, affected herds with high seroprevalence showed a low frequency of the protective allele, except in flocks 3 and 5 of Valle del belìce and Comisana breed, respectively. It is worth noting that sheep with KK genotype were seropositive in the flock 5 of the Comisana breed, characterized by the highest seroprevalence (65%). An infection with a genetically divergent MVV strain that is different from the MVV strain(s) circulating in the other flocks could be the reason for this particular flock’s seroprevalence. Indeed, recent genetic and epidemiological studies, conducted in Italy identified at least three genotypes and great variability of subtypes, even inside the same flock [13,14].

The great majority of the MV serologically positive animals at the breed level resulted in the Valle del Belìce breed, even though there were no positive serologic tests in a Valle del Belìce flock (flock 3) localized in TP province characterized by similar allelic frequencies to other VDB flocks. The farm management system, the history of the animals’ trade, and the environmental conditions in the geographical area in question should be investigated in order to explain this particular case. Although in the present study, association analysis for the Valle del Belìce breed was prevented because the KK genotype frequency was too low, a high MV prevalence came along with a high E allele frequency, which resulted predominantly in agreement with the results of recent studies conducted in German, Iranian, Turkish, and Spanish breeds [22,23,24,26]. In particular, the very low frequency of a protective allele (K35) in the Valle del Belìce reflects the results recently obtained for the Sarda sheep breed (unpublished data). These results suggest that this low frequency of the protective allele (K35), both in Valle del Belìce and Sarda breeds, is probably related to their heritage coming from the farmers’ selection mainly addressed to obtain a highly productive dairy sheep.

In contrast, a more balanced distribution of the protective K35 allele is observed in the other three breeds (Comisana, Barbaresca, and Pinzirita), whose population is now reduced to a few breed groups. The sample size of those endangered breeds (Barbaresca and Pinzirita) and the possible inbreeding within flocks may affect the enhancement of this peculiar biodiversity in the future. Indeed, natural and uncontrolled mating is the standard practice for local farmers with minimal genetic transfer exchange among flock populations, increasing inbreeding [37].

In prospective studies of sheep exposed to MV, the K35 variant has been exhibited to confer a significantly reduced vulnerability to infection compared to the E35 variant [21,36]. However, in natural virus exposure conditions, the reduced susceptibility conferred by the homozygous or heterozygous *TMEM154* genotypes is challenging to quantify. To provide adequate power in the association of *TMEM154* E35K polymorphism with the susceptibility for MV infection, it is necessary to evaluate flocks characterized by a “moderate” level of infection (e.g., not all or most sheep should be positive or negative) and a balanced ratio of sheep carrying the genotypes susceptible (EE/EK) and resistant (KK), but such ideal conditions are challenging to find in the field. For this reason, in the present study, we had to suspend the Valle del Belìce, Barbaresca, and Pinzirita breeds from the association test. Specifically, the VDB breed showed an unbalance of allelic and genotype frequencies (absence of MV-positive sheep with KK genotype), while BAR and PIN breeds showed less than 10% of prevalence of MV-positive sheep. Therefore, a significant association of *TMEM154* E35K with MV serological status of sheep was only found in the total sample set of Sicilian sheep.

In the total sample set, animals carrying one or two copies of allele E35 had a relative risk of 3.8 to be serologically MV positive. This value falls within the relative risk range (from 1.27 to 5.30) reported by Heaton et al. [19] in American sheep breeds, whilst lower values have been shown in German (from 1.39 to 2.26), Iranian (from 0.48 to 2.18), Turkish (3.14), Spanish (from 0.97 to 1.54), and other Italian flocks (1.24) [22,23,24,25,26]. At the breed level, animals carrying the susceptible genotypes (EE/EK) seem to show a greater predisposition to MV infection; however, these results were not significant even if the Comisana breed narrowly missed the significance threshold (*p*-value = 0.08). 

Despite strong evidence for a consistent association between *TMEM154* genotype and susceptibility to MV infection in a wide range of conditions, several authors reported some animals serologically MV positive were carriers of protective genotype [19,22,25]. Several factors may influence the proportion of seropositive sheep with this genotype, such as heterogeneity of breeds, age, and sex of animals, management systems, viral strength, the dose of exposure, route of infection, and presence of different MVV strains or subtypes [19,38,39,40].

Sicilian sheep breeds have also been investigated for their genetic resistance to other diseases. High genetic resistance against scrapie was detected in Valle del Belìce, Comisana, and Pinzirita sheep, with a ranging prevalence from 32.8 to 40.4% of resistant ARR haplotype [41]. Native breeds are often neglected because of the introduction of more productive exotic breeds; nevertheless, they still remain the most sustainable livestock in low-income or marginal rural areas. Furthermore, local breeds represent a precious reservoir of genetic diversity that may be crucial in the face of climate change. Unfortunately, according to the Domestic Animals Diversity-Information System (DAD-IS) FAO database, nowadays, many Sicilian sheep breeds are recognized as endangered or in critical risk status. Therefore, focusing on the maintenance of traits that confer natural resistance against infectious disease may stimulate the conservation of biodiversity and recovery of this genetic potential as suggested by FAO [42] and strongly recommended by the European Green Deal [43].

## 5. Conclusions

Considering the limitations of traditional strategies for the containment of MV infection, a marker-assisted selection to increase the frequency of the protective genotype could represent the most suitable tool to control the disease. Few studies have been conducted in Italy to identify the protective *TMEM154* genotype. Therefore, for the first time, this study reported a genetic investigation of the E35K polymorphism and an association of *TMEM154* genotype with serological MV status in four Sicilian sheep breeds. Comisana, Barbaresca, and Pinzirita breeds showed a good frequency of the protective allele K, whilst the high frequency of risk allele found in the Valle del Belìce breed is coherent with its genetic origin from the Sarda sheep breed. A significant association between *TMEM154* polymorphism to MV illness was found in the sheep population studied in this report. Generally, a high MV prevalence comes along with a high E allele frequency, with some exceptions related to the heterogeneous management system of Sicilian ovine farming. Our data could be helpful in establishing selection programs aimed at controlling and eradicating MV infection in Sicilian sheep farming and could increase interest in breeding Sicilian native breeds, preserving the biodiversity on the island. Today, it is necessary to consider the endangered status of endemic breeds in order to preserve and enhance local genetic variability and biodiversity, including those characteristics linked to disease resistance.

## Figures and Tables

**Table 1 animals-12-01630-t001:** Origin (Sicilian province), breed, number of sampled animals, percentage of serologically MV positive animals for each sampled sheep flock.

Flock	Province *	Breed **	Sampled Animal (*n*)	Serological MV Positive Animals (%)	*TMEM154* Allele Frequencies	
					E	K
1	AG	VDB	61	12 (20%)	0.81	0.19
2	AG	VDB	39	17 (44%)	0.82	0.18
3	TP	VDB	62	0	0.86	0.14
4	PA	VDB	93	45 (48%)	0.97	0.03
5	PA	COM	63	41 (65%)	0.58	0.42
6	EN	COM	13	0	0.77	0.23
7	AG	COM	4	0	0.63	0.38
8	EN	COM	12	0	0.54	0.46
9	AG	BAR	6	0	0.33	0.67
10	AG	BAR	4	0	0.50	0.50
11	CL	BAR	16	1 (6%)	0.56	0.44
12	ME	PIN	19	1 (5%)	0.50	0.50
13	ME	PIN	20	0	0.63	0.38
14	ME	PIN	14	0	0.71	0.29
15	ME	PIN	16	0	0.53	0.47

* Province: Agrigento (AG), Enna (EN), Caltanissetta (CL), Trapani (TP), Palermo (PA), Messina (ME). ** Breeds: Valle del Belìce (VDB), Comisana (COM), Barbaresca (BAR), Pinzirita (PIN).

**Table 2 animals-12-01630-t002:** Primer and probe sequences used for *TMEM154* genotyping and sequencing.

Primers–Probes Sequence (5′-3′)	Amplified Region	Amplicon Size (bp)	Purpose
GTCTCCATGACAAGTCTCAATTTTGT (forward) GCTTAGGGCCTCTGACTCTTCA (reverse) Probe allele K 6FAM-AGGACACA**A**AACTGT- BHQ1 Probe allele E HEX-AGGACACAGAACTGT- BHQ1	Exon 2	117	Detection of nucleotide substitution rs408593969, g.5776842 G > A leading the amino acid substitution E35K using the TaqMan allelic discrimination method [26]
GCTCCATTTATGTTCAATCA (forward) GAGATGGAAGCTGTGTGTTTC (reverse)	Exon 2–3	771	Amplification and sequencing for verification of genotyping results [19]

Bold letters: nucleotide positions leading to allele specific binding of probe.

**Table 3 animals-12-01630-t003:** *TMEM154* genotype and allele frequencies in Sicilian ovine breeds.

Breed	*n*	*TMEM154* Genotype Frequencies			*TMEM154* Allele Frequencies		HWE *p*-Value
		EE	EK	KK	E	K	
VDB	255	200 (0.78)	51 (0.20)	4 (0.02)	0.88	0.12	0.719
COM	92	34 (0.37)	43 (0.47)	15 (0.16)	0.60	0.40	0.821
BAR	26	6 (0.23)	14 (0.54)	6 (0.23)	0.50	0.50	0.695
PIN	69	27 (0.39)	27 (0.39)	15 (0.22)	0.59	0.41	0.109
Total	442	267 (0.60)	135 (0.31)	40 (0.09)	0.76	0.24	0.000

**Table 4 animals-12-01630-t004:** Association analysis between *TMEM154* genotype and serological MV status in Sicilian ovine breeds.

Sheep Subset (*n* Sheep)	MV Status (*n* Sheep)	*TMEM154* Allele Frequencies		*TMEM154* Genotype Frequencies			Chi-Square *p-*Value	RR	95% CI	*p-*Value
		E	K	EE	EK	KK				
Total (442)	Positive (117)	0.83 (195)	0.17 (39)	0.70 (81)	0.28 (33)	0.02 (3)	0.008	3.8	1.3–11.4	0.018
	Negative (325)	0.73 (474)	0.27 (176)	0.60 (186)	0.30 (102)	0.10 (37)				

CI: confidence interval.

## Data Availability

The data that support the findings of this study are publicly available on following https://doi.org/10.6084/m9.figshare.19525327.v1 (accessed on 15 May 2022).

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
