# Peer review of "Alternative Molecular Tools for the Fight against Infectious Diseases of Small Ruminants: Native Sicilian Sheep Breeds and Maedi-Visna Genetic Susceptibility"

_animals, 2022, doi:10.3390/ani12131630_

Round 1
Reviewer 1 Report
Review Report Animals-MDPI
Brief summary
This manuscript focused on application of alternative biomolecular tools for the evaluation of genetic resistance to SRLV (Maedi-Visna) in Sicilian sheep breeds. In particular, authors assessed, by genotyping or better allelic discrimination, the presence of the polymorphism SNP E35K within the TMEM154 gene, notably the genetic marker key in the fight against this infectious disease. On the other hand, with sequencing novel SNPs were investigated but not detected.The novelty and the strenght point of this work is particularly the performing for the first time of susceptibility study in local Sicilian sheep breeds, reservoir of “ancient” genetic traits important also for conferring natural resistance to different pathogens.
Broad comments
This manuscript focused on application of alternative bio-molecular tools for the evaluation of genetic resistance to SRLV (Maedi-Visna) in Sicilian sheep breeds. In particular, authors assessed, by genotyping or better allelic discrimination, the presence of the polymorphism (SNP) E35K within the TMEM154 gene, notably the genetic marker key in the fight against this infectious disease. On the other hand, novel SNPs located on the amplified gene were investigated by Sanger sequencing but not detected. The novelty and the strenght point of this work is particularly the performing for the first time of susceptibility study in local Sicilian sheep breeds, reservoir of “ancient” genetic traits important also for conferring natural resistance and adaptability to different pathogens and to climate change.
The work is interesting because fairly current to be published (as communication but also as original article); study design is valid as well as the sampling size total and for each defined breed. I suggest to implement the sequencing approach, particularly focusing and detailing the platform and software used for multiple-alignment of the sequences.
Furthermore I suggest to control again English grammar and some passages formulated in English language.
Title: I suggest to specify in the title the adopted tools and the species object of the study, in particular: “Alternative molecular tools for the fight against disease of small ruminants: native Sicilian sheep breeds and Maedi-Visna genetic susceptibility”.
Simple summary
Line 24: I suggest to specify that other target genes were studied in this field, among them TLR9, CCR5, Myd88 genes, that play important role in immunity and in genetic “resilience” to SRLVs,
Abstract
Line 40: please, remove the term “ frequency” because redundant.
Introduction
This section is well detailed, but some updated references particularly about SRLV genetic characterization and phylogenetic studies conducted also in your country (Italy) are missing.
Line 92: please correct “an unique” into “a unique”
Materials and methods
Table 1: please, indicate breed acronyms also in the Table legend.
2.3 DNA extraction and TMEM154 genotyping
I suggest to implement the sequencing approach, particularly focusing and detailing the platform and software used for multiple-alignment of the sequences.
Results
Line 203: please reformulate the sentence into “No new SNPs were found in the target TMEM154 region, assessed by sequencing analysis.
Line 210: please, correct K35 into E35.
Lines 216 and following: this passage is not clear. Please explain better this concept, referring to the examined and investigated breeds. For example, Valle del Belice breed is characterized by an unbalance of allelic and genotype frequencies but the number of sheep positive to sierological assays is quite high, differently from Barbaresca and Pinzirita.
Lines 230-233: please, clarify this paragraph. Relatively to this statement, is the heterozygous EK genotype known as risk factor in disease predisposition/susceptibility? I suppose so. Please, introduce and add supporting literature also in discussion section.
Discussion
Lines 267-269: I suggest corroborating this “speculation” also indicate the even lower protective genotype frequency detected in the investigated population (0.02, n=3).
Lines 296-298: please, better explain this concept and this reported statement “We had suspend the Valle del Belice, Barbaresca and Pinzirita breeds from the association test”
Line 316: probably, you wanted to refer to reference 38. Please change and control the other references.
Lines 317-318: this statement is incomplete.
References
Please check the reported references also in accordance with the format required by “Animals-MDPI.
Reviewer 2 Report
Over the manuscript is well written, but some improvements in data analyse and result presentation are needed.
The figures in Table 1 show considerable variation in the percentage of serological MV positives within breeds. This, together with the results obtained from this study, suggest the susceptibility to MV may be related to TMEM154 but not breed. Therefore the analyses of genotype and MV positive cases for individual breeds may be misleading.
I would suggest: 1) include one column in Table 1 showing the percentage of sheep having one or two copies of the E allele; 2) Tables 3 and 4 can be deleted; 3) the COM breed subset in Table 5 can be removed; 4) if possible, look at the association for one copy vs two copies vs zero copy.
Reviewer 3 Report
animals-1754748
Alternative tools for the fight against infectious diseases of small ruminants: native Sicilian breeds and Maedi-Visna susceptibility
S. Tumino et al.
General comments:
In this manuscript, the authors investigate the association of TMEM154 E35K allele frequencies with MV susceptibility in native Sicilian sheep breeds. This is an important contribution to the Sicily region which produce high quality sheep milk. The manuscript is well written, and the results are clearly presented. Specific points to be addressed are as follows.
Specific comments:
L189-194: Specify that these statements are in regards to Table 4. Otherwise, it’s difficult to follow which result (table) is explained. In addition, please reconsider the order of the tables.
L210: It should be E35, instead of K35.
L279-281: Please show the citation.
L301-302: “~whilst lower values have been~”. Please show the specific values.
L308: “animals” should be written as “some animals”, and remove “despite”.
L309-312: Table 1 shows that for the VDB breed, interestingly, there were no positive serologic tests, despite the fact that TP province had the similar number of samples and positive rate as the other two provinces (AG and PA). In these statements the authors list a variety of factors, but what is the explanation for this?
Round 2
Reviewer 2 Report
The reason that I asked the authors to include allele frequencies in Table 1 is because this information may give some kind of indication on whether there is any association pattern between allele frequencies and the serological MV positive percentages. It is okay if the authors insist not to do this.
However, for the association analysis shown in Table 3, the authors should also compare EE vs EK/KK. If it has been confirmed that both the EE and EK genotypes are susceptible to the infection while only the homozygous KK genotype confers resistance, why did the authors conduct this study? The authors also need to discuss this results in the discussion section.
Round 3
Reviewer 2 Report
The revised manuscript looks better. The prevalence of MV illness appeared to be associated with the frequency of K allele in all flocks of sheep, with the exception in flocks 3 and 5. The authors should discuss the potential reasons for this. Was this related to environmental conditions or farm management?
